# Making Hard Problems Easier with Custom Data Distributions and Loss Regularization: A Case Study in Modular Arithmetic

**Eshika Saxena** [* 1]  **Alberto Alfarano** [* 1]  **Emily Wenger** [† 1 2]  **Kristin Lauter** [† 1]

## Abstract

Recent work showed that ML-based attacks on Learning with Errors (LWE), a hard problem used in post-quantum cryptography, outperform classical algebraic attacks in certain settings. Although promising, ML attacks struggle to scale to more complex LWE settings. Prior work connected this issue to the difficulty of training ML models to do modular arithmetic, a core feature of the LWE problem. To address this, we develop techniques that significantly boost the performance of ML models on modular arithmetic tasks—enabling the models to sum up to $N = 128$ elements modulo $q \leq 974269$. Our core innovation is the use of custom training data distributions and a carefully designed loss function that better represents the problem structure. We apply an initial proof of concept of our techniques to LWE specifically and find that they allow recovery of 2x harder secrets than prior work. Our techniques also help ML models learn other well-studied problems better, including copy, associative recall, and parity, motivating further study.

## 1. Introduction

Modular arithmetic is a key component of many cryptographic hard problems, including Learning with Errors (LWE), which is the basis for several newly standardized post-quantum cryptosystems (Chen et al., 2022). The LWE problem involves determining a secret vector given two pieces of information: $\mathbf{A} \in \mathbb{Z}_q^{m \times n}$, a matrix sampled uniformly at random mod $q$, and $\boldsymbol{b} = \mathbf{A} \cdot \boldsymbol{s} + \boldsymbol{e} \in \mathbb{Z}_q^m$, the noisy dot product of $\mathbf{A}$ with the secret vector $\boldsymbol{s}$ mod $q$. When $\boldsymbol{s}$ is binary, which often happens in practical applications of LWE, computing $\boldsymbol{b}$ from $\mathbf{A}$ requires computing a subset sum mod $q$, which is a modular addition problem.

---
*Equal contribution † Equal contribution [1]Meta AI [2]Duke University. Correspondence to: Eshika Saxena <eshika@meta.com>.

*Proceedings of the 42$^{nd}$ International Conference on Machine Learning*, Vancouver, Canada. PMLR 267, 2025. Copyright 2025 by the author(s).

Recent work has shown that machine learning (ML) models can, under certain conditions, be used to break LWE by recovering the secret (Wenger et al., 2022; Li et al., 2023a;b; Stevens et al., 2024). The attack trains ML models to predict $\boldsymbol{b}$ given $\mathbf{A}$, which requires learning to compute the (noisy) modular arithmetic operation $\mathbf{A} \cdot \boldsymbol{s} \pmod{q}$. If models predict $\boldsymbol{b}$ with even low accuracy, $\boldsymbol{s}$ can be recovered.

However, ML attacks struggle to recover LWE secrets with many nonzero entries, limiting their efficacy. Prior work links this struggle to ML models' inability to learn modular arithmetic at scale, since recovering LWE secrets with more nonzero entries requires summing more elements mod $q$ (Wenger et al., 2024). Given the widespread adoption of LWE-based cryptosystems, exploring the potential of these nascent ML attacks—which can exploit statistical dependencies algebraic attacks may miss—is critical to improve understanding of LWE's practical security.

ML models' struggle to perform modular arithmetic extends beyond cryptanalysis. Numerous works have documented the poor performance of ML models on modular arithmetic problems with many terms and/or large prime moduli (Palamas, 2017; Gromov, 2023; Abbe et al., 2023; Mohamadi et al., 2024; Doshi et al., 2024). This is surprising because ML models can learn other complex math tasks such as symbolic regression, linear algebra, discovering Lyapunov functions, computing Gröbner bases, polynomial simplification, and computing the greatest common divisor (Charton et al., 2021; Charton, 2022; Alfarano et al., 2024; Kera et al., 2024; Agarwal et al., 2021; Charton, 2024). Modular arithmetic, on its face, seems easier, but scalable ML solutions remain elusive.

Thus, our goal is simple: *train ML models that can sum many elements mod $q$*. In considering this challenge, we make two key observations. First, modular arithmetic problems lie along a gradient of difficulty based on how many times the sum "wraps" around the modulus. Prior work observed that models learn better from examples that wrap fewer times around the modulus $q$, e.g. $\sum_{i=1}^{n} a_i \leq 2q$ (Wenger et al., 2024). Second, the geometry of the modular field disrupts traditional notions of gradient descent, as 0 and $q$ are close (Shalev-Shwartz et al., 2017). Prior work proposed an angular embedding for model input/outputs that

better represented this geometry (Stevens et al., 2024), but this structure was not explicitly used during optimization.

**Our contribution.** Building on these observations, we hypothesize that two changes to the training process—using custom training data distributions and problem-specific loss regularization—may help models better learn modular arithmetic. The use of custom data distributions is inspired by recent prior work that demonstrated the efficacy of custom data in related domains (Charton & Kempe; Abbe et al., 2023). Crafting training data that simultaneously exposes models to easy and harder versions of modular arithmetic (e.g. elements that wrap fewer or more times around $q$) could help the model unlock the structure of the problem. Loss regularization is widely used in ML training, but could be adapted to better represent modular arithmetic specifically, as noted by prior work (Gromov, 2023; Jelassi et al., 2023). Building on this, we:

- Propose a *novel training data distribution* that includes both easy and hard variants of the modular arithmetic problem. This distribution—implemented as a function over training elements rather than a curriculum—smoothly adjusts the hardness of elements seen by the model, enabling efficient and effective learning.

- Design a *custom loss function* with a penalty term specific to modular arithmetic. This function discourages model convergence at local, unhelpful minima, enabling discovery of global optima appropriate to our problem setting.

**Key results.** Our methods enable ML models to perform modular addition for a variety of $N$ and $q$, up to $N = 128$ and $q = 974269$. This *significantly* outperforms prior work, which summed $N \leq 6$ elements mod $q \leq 1000$. When we apply our methods to the LWE problem specifically, we find we can recover 2x harder secrets than prior work.

Although our motivation is LWE-specific, our methods could aid learning on other problems as well. In §4.2, we highlight several interesting findings about how models learn modular arithmetic, which yield insights that could aid learning in other settings. We also apply our methods to other well-studied problems (copy, associative recall, and parity) and find they enable models to learn better, demonstrating their potential to generalize. Our code is available at: https://github.com/facebookresearch/arithmetic.

## 2. Related Work

**ML for modular arithmetic.** Previous work has investigated whether ML models can learn modular arithmetic operations (Palamas, 2017; Lauter et al., 2024; Gromov, 2023; Abbe et al., 2023; Mohamadi et al., 2024; Doshi et al.,

2024). Table 1 summarizes the best prior results on modular addition. The best existing methods train models that sum $N \leq 6$ elements for moduli up to $q = 1000$.

Prior work has laid groundwork for analytically understanding how models learn modular arithmetic (Gromov, 2023; Doshi et al., 2024; Nanda et al., 2023). Modular arithmetic is a common task used to study "grokking" as the model suddenly transitions from memorization to generalization on this task. Some works go on to show that ML models are able to inherently learn the problem by encoding the modular arithmetic structure into the model weights (Nanda et al., 2023; Doshi et al., 2024). However, prior work consistently shows (Table 1) that models struggle to learn modular arithmetic as the number of summed terms $N$ and modulus $q$ increase, motivating our search for new learning methods.

*Table 1.* **Summary of prior work on ML-enabled modular addition**. Best $N$ and $q$ are **bold**.

| Paper | # Terms ($N$) | Mod ($q$) | % Accuracy |
|---|---|---|---|
| Nanda et al. (2023) Transformer | 2 | 53, 109, 113, 401 | 100 |
| Mohamadi et al. (2024) 2-layer MLP | 2 | **433** | 100 |
| Doshi et al. (2024) 2-layer MLP | **6** | 11, 23 | 97.1 |
| Gromov (2023) 2-layer MLP | 2 | 97 | 100 |
| Jelassi et al. (2023) Encoder-only transformer | 2 | 100, **1000** | 73 |
| Abbe et al. (2023) 4-layer MLP | 2 | 2 | 100 |

**Custom training data distributions.** Some prior work has explored how changing the training data distribution affects performance on arithmetic tasks. For example, Charton & Kempe show that repeating training examples improves transformer performance on arithmetic tasks like modular multiplication, greatest common divisor, and eigenvalue calculation. Mohamadi et al. (2024) briefly investigated the effect of training data on modular arithmetic and found that models need to be trained on a constant fraction of all possible modular arithmetic behaviors for a given $N$ and $q$ to generalize. These works motivate further investigation into how the distribution of training data can affect model performance on arithmetic tasks.

Related work has investigated the impact of changing the training data distribution over time in a predetermined pattern, a technique known as curriculum learning (CL) (Bengio et al., 2009). Abbe et al. (2023) show that the parity function can be learned via curriculum learning strategies. We note that our approach is different than CL because we do not change the data distribution over time, so our approach requires less tuning than CL. Our experiments

show that CL is highly sensitive to the curriculum parameters, whereas our approach is simpler and ensures consistent convergence (see Table 4).

**Effect of loss functions.** Several works attribute models' failure to learn harder modular addition problems to the complexity of the loss space (Gromov, 2023; Jelassi et al., 2023). Shalev-Shwartz et al. (2017) highlight the limitations of gradient-based methods on different problems, including learning random parities. These motivate adding loss regularization based on the problem to overcome these challenges.

## 3. Methodology

Following prior work (Jelassi et al., 2023), we train encoder-only transformer models to add $N$ elements mod $q$ (fixed $N$ and $q$ for each model). We also leverage angular embeddings proposed by Stevens et al. (2024), which map input and output elements mod 0 to points on the unit circle. These better represent the structure of modular arithmetic since 0 and $2\pi$, which corresponds to $q$, are close under this geometry. This section describes our key innovations to the training pipeline, custom training data and loss regularization, then gives an overview of our end-to-end training procedure and evaluation metrics.

### 3.1. Innovation 1: Augmenting Training Data with Sparse Vectors.

Most prior work on ML for modular arithmetic trains models using randomly generated $(\boldsymbol{a}, b)$ pairs, where $\boldsymbol{a}$ is drawn uniformly at random from $\mathbb{Z}_q^N$, i.e. $\boldsymbol{a}$ consists of elements $[a_1, a_2, \ldots, a_N]$, $a_i \in \mathbb{Z}_q$ and $b = \sum_{i=1}^{N} a_i \pmod{q}$ (Jelassi et al., 2023; Doshi et al., 2024). Based on observations about the importance of training data diversity from Wenger et al. (2024) and Mohamadi et al. (2024), models learn better when they see more examples of "simpler" versions of the target operation. Seeing these simplified problems may help models understand the modular arithmetic structure and learn better. Thus, we propose adding additional *sparse* vectors to the training data, in which more coordinates of $\boldsymbol{a}$ are 0, alongside more general vectors.

**Data generation approach.** We selectively add sparse vectors to the training data by employing a sampling distribution $f$ in our data generation procedure, which controls the number of nonzero elements in a training data element $a$. To generate $\boldsymbol{a}$, we first sample from $f$ to get a number $n \in [1, N]$, the number of nonzero elements in $\boldsymbol{a}$. We then sample the nonzero elements of $\boldsymbol{a}$ from $\mathbb{Z}_q^n$, pad this with a zero string of length $N - n$, and shuffle the resulting vector to ensure 0s are randomly distributed.

We experiment with two $f$ distributions: $f_{\text{uni}}(z) = \frac{1}{N}$ (i.e.

uniform density) and $f_{\text{inv\_sqrt}}(z) \propto \frac{1}{\sqrt{N-z+1}}$ (similar to the distribution chosen by Allen-Zhu (2025)) where $\propto$ means the functions are rescaled by a constant such that the sum of $f$ over all $z$ in its domain equals 1. We compare these to a baseline of $f_{\text{default}}$, which is the PDF of the number of zeros in $\boldsymbol{a}$ when $\boldsymbol{a}$ is drawn uniformly from $\mathbb{Z}_q^N$. Most prior work on ML for modular arithmetic used $f_{\text{default}}$ for training data generation, as noted previously.

**Effect of $f$ on training data.** Figure 1 shows the sparsity of training data elements created using these sampling strategies with $N = 16$ and $q = 257$. Note that $f_{\text{inv\_sqrt}}$ is biased towards creating "harder" elements with few 0s but still allows some easier examples to exist, whereas $f_{\text{default}}$ rarely creates examples with many zeros, meaning that the model must learn only from hard examples. During testing, we evaluate models on examples drawn uniformly at random from $\mathbb{Z}_q^N$, since general modular addition is the target task.

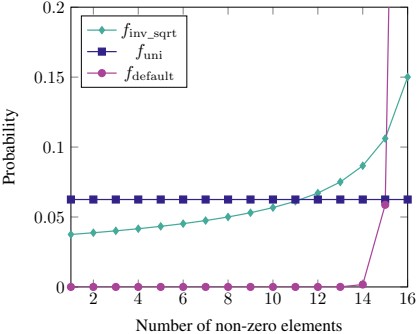

*Figure 1.* **Probability of number of non-zero elements in each training data element when $N = 16$ and $q = 257$ for $f_{\text{inv\_sqrt}}$, $f_{\text{uni}}$ and $f_{\text{default}}$ sampling distributions.**

### 3.2. Innovation 2: Loss Regularization to Avoid Model Collapse.

Following Stevens et al. (2024), we use an angular embedding to represent transformer inputs and outputs. Practically, the embedding encodes an integer $t \in \mathbb{Z}_q$ as an angle $\phi = 2\pi \frac{t}{q}$ and then as a point $(\cos(\phi), \sin(\phi)) \in \mathbb{R}^2$ on the unit circle. Based on this, we initially chose an MSE loss for training. However, we observed that in harder settings the model would often converge to bad local minima like the origin. To address this issue, we use a custom loss function during training that combines mean squared error (MSE) loss with an extra regularization term. Given a prediction of the form $(x', y')$ and ground truth $(x = \cos\phi, y = \sin\phi)$, this loss takes the form:

$$\ell = \alpha \left( x'^2 + y'^2 + \frac{1}{x'^2 + y'^2} \right)$$
$$+ \left( (x - x')^2 + (y - y')^2 \right), \quad \alpha = 10^{-4}$$

The first term penalizes the model for predicting the origin by driving the loss to infinity if $x' = 0, y' = 0$. It also encourages the model to predict $(x', y')$ on the unit circle (the first term is minimized with $x'^2 + y'^2 = 1$). The second term is the standard MSE loss. After some training $x'$ and $y'$ are close to the unit circle, so we can approximate $x'$ and $y'$ as $\cos\phi'$ and $\sin\phi'$. Under this condition, the MSE loss function component becomes:

$$\begin{aligned}\ell &\approx (\cos\phi - \cos\phi')^2 + (\sin\phi - \sin\phi')^2 \\ &= 2 - 2\cos\phi\cos\phi' - 2\sin\phi\sin\phi' \\ &= 2 - 2\cos(\phi - \phi')\end{aligned}$$

This loss component will be minimized when $\cos(\phi - \phi') \approx 1$, which occurs at $\phi - \phi' = 0$ and $\phi - \phi' = 2\pi$. In the modular arithmetic setting, we want 0 and $2\pi$ to be understood as "close" in the loss space, so this loss term correctly describes the desired behavior.

### 3.3. Model Training and Evaluation

We implement the proposed changes and train encoder-only transformers to sum $N$ elements mod $q$, using the following settings and evaluation metrics.

**Parameter selection.** We experiment with $N = \{16, 32, 64, 128\}$ to identify trends as $N$ increases. Because we are interested in the LWE application, we use prime moduli, which are commonly used in that setting. We use $q = \{257, 3329, 42899, 974269\}$, including one ($q = 3329$) used in a standardized LWE-based cryptosystem, CRYSTALS-KYBER (Avanzi et al., 2021). We also tested with non-prime modulus $q = 2^{16}$ and obtained similar results, as shown in Appendix A. We select $N = 16, q = 257$ as our base case because the sample space is large enough to ensure models generalize.

**Training procedure.** All our experiments were implemented in Python with Pytorch. We train the transformer models with a hidden dimension of 256, 4 attention heads, and 4 encoding layers on batches of 250 examples, using the Adam optimizer (Kingma & Ba, 2015) with a learning rate of $3 \cdot 10^{-5}$, an initial linear warm-up phase of 1,000 optimization steps, and cosine scheduling. All experiments run on 1 V100 GPU with 32 GB of memory. The models were trained with $10M$ distinct samples for a total $100M$ samples. Training time is around 3 hours.

**Evaluation metrics.** We generate a held-out test set $\mathcal{D}_{\text{test}}$ of size 100,000 that is distinct from the training set and contains examples drawn uniformly from $\mathbb{Z}_q^N$. To evaluate model performance on $\mathcal{D}_{\text{test}}$, we take the final hidden state of the transformer and pass it through a linear layer to produce an output of the form $(x', y')$. We project this point onto the unit circle, producing $(\cos\phi', \sin\phi') = (\cos\frac{2\pi}{q}s'\sin\frac{2\pi}{q}s')$. The model prediction is then compared

against the ground truth of $(\cos\frac{2\pi}{q}s, \sin\frac{2\pi}{q}s)$.

To get a complete picture of model performance, we compute the following metrics: Mean Squared Error (MSE) of angle predictions and % accuracy with a margin of error ($\tau$) relative to $q$. MSE help us to evaluate the model's performance in terms of closeness between the predicted and ground truth angles (MSE). $\tau$-accuracy enables us to measure whether the model learns the approximate function behavior, even if exact accuracy is low. The formulae for these metrics are:

$$\text{MSE} = \frac{1}{|\mathcal{D}|}\sum_{x\in\mathcal{D}}\left((\cos\phi - \cos\phi')^2 + (\sin\phi - \sin\phi')^2\right)$$

$$\tau\text{-accuracy} = \frac{1}{|\mathcal{D}|}\sum_{x\in\mathcal{D}}\mathbb{1}_{\|s'-s\|\leq\tau q}$$

## 4. Results and Further Exploration

### 4.1. Key Results

**Best results on modular addition.** Our methods enable models to learn modular addition of up to $N = 128$ elements mod $q$. We present best results across a range of $N$ and $q$ values in Table 2. Overall, the MSE is near 0 across $N$ and $q$, showing that the model converges and learns well. Notably, $\tau = 0.5\%$ accuracy is almost perfect for all models. This means that in almost all cases, an "incorrect" model prediction is still within $0.5\%$ of $q$.

**Comparison to baseline.** We compare our results to a baseline encoder-only transformer with standard MSE loss and the $f_{\text{default}}$ distribution, as is used in some of the prior work (Jelassi et al., 2023). With the baseline approach, we observe $1.3\%$ $\tau = 0.5\%$-accuracy on $N = 16$, $q = 257$ data (our base case) with the same number of training data samples as we used. In other words, the model does not learn the task at all. In comparison, our methods achieve $99.8\%$ on the same problem. Unlike Gromov (2023) and Doshi et al. (2024), we do not observe grokking in our models because we use a very small fraction of data from the possible sample space ($2.76 \cdot 10^{-31}$ when $N = 16$ and $q = 257$). As such, our models gradually learn with a standard training loss behavior and do not overfit.

### 4.2. Deeper Exploration of Our Methods

Beyond simply improving ML performance on modular arithmetic, our methods offer interesting lessons on the learning process. Here, we present results that validate our approach and give more general insight about how models learn modular arithmetic.

**Models learn easy examples before hard ones.** A key motivation for our approach is that models should initially

*Table 2.* **Our methods perform consistently well adding** $N \in$ $[16, 32, 64, 128]$ **elements** mod $q \in [257, 3329, 42899, 974269]$**.** All metrics are computed on a held out test set. MSE is mean squared error, $\tau = 0.5\%$ Accuracy is percentage of predictions within $0.005q$ of right answer and $\tau = 1\%$ Accuracy is percentage of predictions within $0.01q$ of right answer (see §3). The models perform with consistently low MSE and very high $\tau$-accuracy.

| # Terms ($N$) | Mod ($q$) | MSE | $\tau = 0.5\%$ Accuracy | $\tau = 1\%$ Accuracy |
|---|---|---|---|---|
| 16 | 257 | 0.00 | 99.8% | 100.0% |
| | 3329 | 0.00 | 99.7% | 100.0% |
| | 42899 | 0.00 | 99.7% | 100.0% |
| | 974269 | 0.00 | 99.7% | 100.0% |
| 32 | 257 | 0.00 | 99.5% | 100.0% |
| | 3329 | 0.00 | 99.4% | 100.0% |
| | 42899 | 0.00 | 99.4% | 100.0% |
| | 974269 | 0.00 | 99.5% | 100.0% |
| 64 | 257 | 0.01 | 98.9% | 99.4% |
| | 3329 | 0.01 | 97.4% | 99.4% |
| | 42899 | 0.01 | 97.4% | 99.4% |
| | 974269 | 0.01 | 98.2% | 99.4% |
| 128 | 257 | 0.04 | 96.1% | 98.2% |
| | 3329 | 0.04 | 92.9% | 98.0% |
| | 42899 | 0.05 | 94.1% | 97.9% |
| | 974269 | 0.04 | 93.3% | 97.4% |

learn from the sparse training examples before learning the full task. To validate that this occurs, we train a model on $N = 64$, $q = 257$ and monitor its performance on a dataset $\mathcal{D}_{\text{val}}$ drawn from the same distribution as $\mathcal{D}_{\text{train}}$. Figure 2 shows model accuracy on samples with 1 to 64 nonzero elements over the training epochs. Here, we see that the model initially performs better on sparse examples (e.g. 1 non-zero element) and then becomes accurate on more complex examples in later epochs. This suggests that these models first learn simpler sums and build on that knowledge to learn more complex sums, supporting our use of sparsity sampling in creating training data.

**Sparse data elements are critical for learning.** As described in §3, we construct more diverse training datasets by sampling elements with sparsity defined by a PDF $f$. Here, we explore how different PDFs ($f_{\text{default}}$, $f_{\text{inv\_sqrt}}$, and $f_{\text{uni}}$, see §3) affect model performance. We report two metrics: $\tau = 0.5\%$ accuracy of models and the Kullback–Leibler (KL) divergence between the training and testing datasets. KL divergence quantifies the similarity between training dataset $\mathcal{D}_{\text{train}}$, constructed using function $f$, and $\mathcal{D}_{\text{test}}$, sampled from the set $\mathbb{Z}_q^N$ uniformly at random, i.e. $f_{\text{default}}$. As Table 3 shows, the accuracy difference between models trained with the default sampling ($f_{\text{default}}$) and any other distribution $f$ is stark. The exact same architecture has negligible accuracy if we do not modify the training dataset sparsity distribution and achieves over $90\%$ when we do.

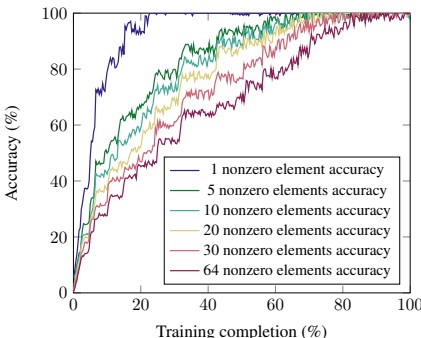

*Figure 2.* **The model learns to sum fewer nonzero elements earlier than more complex examples.** Model accuracy ($N = 64$, $q = 257$) after each epoch on unseen test set stratified by number of nonzero elements. As the number of nonzero elements increases, it takes longer to reach perfect accuracy.

This strongly indicates that these models need to see sparse training examples to generalize.

*Table 3.* **Sampling the training data from** $f_{\text{inv\_sqrt}}$ **produces the best accuracy results across** $N$ **and** $q$**.** $\tau = 0.5\%$ Accuracy is percentage of predictions within $0.005q$ of right answer (see §3), KL divergence is the level of similarity between the training and testing datasets. With default sampling $f_{\text{default}}$, the model does not learn at all. Distributions with a KL divergence that is not too high or too low enable the model to perform best.

| # Terms ($N$) | Mod ($q$) | Training Data $f$ | $\tau = 0.5\%$ Accuracy | KL divergence |
|---|---|---|---|---|
| 16 | 257 | $f_{\text{default}}$ | 1.2% | 0.0 |
| | | $f_{\text{inv\_sqrt}}$ | **99.8%** | 25.2 |
| | | $f_{\text{uni}}$ | 99.7% | 35.4 |
| 32 | 257 | $f_{\text{default}}$ | 1.3% | 0.0 |
| | | $f_{\text{inv\_sqrt}}$ | **99.5%** | 49.8 |
| | | $f_{\text{uni}}$ | 98.9% | 71.5 |
| 64 | 257 | $f_{\text{default}}$ | 1.3% | 0.0 |
| | | $f_{\text{inv\_sqrt}}$ | **98.9%** | 98.1 |
| | | $f_{\text{uni}}$ | 95.3% | 144.0 |
| 128 | 257 | $f_{\text{default}}$ | 1.3% | 0.0 |
| | | $f_{\text{inv\_sqrt}}$ | **96.1%** | 193.7 |
| | | $f_{\text{uni}}$ | 92.7% | 289.5 |

**KL divergence** $\mathcal{D}_{\text{train}}/\mathcal{D}_{\text{test}}$ **impacts accuracy.** We observe that models trained using $f$ that produce very low ($\approx 0$) or very high $\mathcal{D}_{\text{train}}/\mathcal{D}_{\text{test}}$ KL divergence generalize worse than $f$ with mid-range KL divergence. Models trained with the default $f_{\text{default}}$ distribution have 0 $\mathcal{D}_{\text{train}}/\mathcal{D}_{\text{test}}$ KL divergence, since the train/test distributions are almost identical, and model accuracy is $0\%$. On the other hand, the uniform sparsity function $f_{\text{uni}}$ diverges too far from the test distribution (there are too many "simple" samples in the $f_{\text{uni}}$ distribution), resulting in lower accuracy. A "Goldilocks" KL divergence is needed, and we find that distributions with

fewer sparse training elements, like $f_{\mathrm{inv\_sqrt}}$, work best.

**Our data distribution produces more consistent results than curriculum learning.** We explore how our approach compares to standard CL since both involve improving model learning by showing the model easier versions of the task. However, CL requires (a) defining phases of the curriculum by ranking the difficulty of each samples and deciding when to switch phases, (b) determining how to adjust the weight decay and learning rate parameters, (c) defining the data mixture across different phases. Our approach instead fixes these parameters during training by implicitly encoding curriculum "phases" via our data distribution, resulting in a more static but easier-to-train method.

For comparison, we implement CL as follows. First, train the model for a fixed horizon on easier samples only, then train on the entire dataset, as in Abbe et al. (2023). Table 4 indicates that CL results in noisy outcomes, especially with large $N$, when evaluating *average* accuracy and variance over 8 trials. Conversely, our method, which fixes the data distribution during training, ensures convergence. We also explored alternative curriculum designs (e.g. training on easier samples until training loss reaches a certain threshold) and conducted extensive hyperparameter tuning on weight decay, learning rate, alternative difficulty ranking, and data mixture. These alternatives yield results similar to those in Table 4. We provide the specific parameter sets tested in Appendix F and report the best results in Table 4. Ultimately, while it may be possible to design an optimal curriculum, our approach is simpler and task agnostic (see §6).

*Table 4.* **Our approach is more consistent and stable than CL, especially for larger $N$.** We chose the best performing curriculum to compare to our approach. We report the *average* $\tau = 0.5\%$ accuracy (see §3) and the variance across 8 trials.

| # Terms ($N$) | Mod ($q$) | $\tau = 0.5\%$ Accuracy | |
| --- | --- | --- | --- |
| | | Our approach | CL |
| 16 | 257 | $99.7\% \pm 0.0\%$ | $99.7\% \pm 0.0\%$ |
| | 3329 | $99.6\% \pm 0.0\%$ | $99.5\% \pm 0.0\%$ |
| | 42899 | $99.6\% \pm 0.1\%$ | $99.4\% \pm 0.0\%$ |
| | 974269 | $99.6\% \pm 0.0\%$ | $99.4\% \pm 0.0\%$ |
| 32 | 257 | $99.3\% \pm 0.1\%$ | $98.8\% \pm 0.4\%$ |
| | 3329 | $99.2\% \pm 0.1\%$ | $98.0\% \pm 0.5\%$ |
| | 42899 | $99.2\% \pm 0.1\%$ | $97.8\% \pm 1.2\%$ |
| | 974269 | $99.4\% \pm 0.1\%$ | $98.5\% \pm 0.4\%$ |
| 64 | 257 | $98.5\% \pm 0.3\%$ | $95.7\% \pm 2.8\%$ |
| | 3329 | $97.3\% \pm 0.1\%$ | $91.2\% \pm 2.5\%$ |
| | 42899 | $97.3\% \pm 0.2\%$ | $91.3\% \pm 0.9\%$ |
| | 974269 | $98.0\% \pm 0.2\%$ | $93.9\% \pm 3.2\%$ |
| 128 | 257 | $95.8\% \pm 0.5\%$ | $85.1\% \pm 10.3\%$ |
| | 3329 | $92.2\% \pm 0.7\%$ | $82.2\% \pm 8.8\%$ |
| | 42899 | $92.9\% \pm 0.8\%$ | $80.0\% \pm 12.2\%$ |
| | 974269 | $92.1\% \pm 0.4\%$ | $77.3\% \pm 7.5\%$ |

**The data distribution approach is agnostic of the element chosen for sparsity.** Regarding choosing to include 0 as the element for sparsity, we actually found that including a random but fixed number instead of 0 yields similar results. It is not the number itself but instead the shift in distribution that leads to performance improvements. The key is changing the KL divergence between the train and test sets, which is irrespective of the element chosen for sparsity.

To prove this, we run an experiment where we substitute the 0 with an arbitrary integer $K$ and shift the distribution by sampling more $K$s compared to all remaining elements in $Z_p^N$. We used a random $K$ for each different $q$ to ensure $K$ is not a factor. We report the results in Table 5.

*Table 5.* **Our approach works well consistently across different integers used for sparsity.** We train the models with the best data parameters from §4 and include a random number $K$ instead of 0 in the data to make the examples sparse. $\tau = 1\%$ Accuracy is percentage of predictions within $0.01q$ of right answer (see §3)

| # Terms ($N$) | Mod ($q$) | $K$ | $\tau = 1\%$ Accuracy |
| --- | --- | --- | --- |
| 32 | 257 | 160 | 100% |
| | 3329 | 3176 | 100% |
| | 42899 | 24606 | 100% |
| | 974269 | 79062 | 100% |
| 64 | 257 | 160 | 99.2% |
| | 3329 | 3176 | 99.3% |
| | 42899 | 24606 | 99.4% |
| | 974269 | 79062 | 99.3% |
| 128 | 257 | 160 | 97.9% |
| | 3329 | 3176 | 98.3% |
| | 42899 | 24606 | 97.9% |
| | 974269 | 79062 | 97.9% |

**Repeating examples helps.** In applications like LWE cryptanalysis, the amount of training data may be limited, so we consider whether models can learn from fewer samples. Interestingly, recent work shows that it may be beneficial for models to encounter samples multiple times during training, suggesting that limited training data may not be as problematic as anticipated (Charton & Kempe). To test whether repeated examples help in our setting, we train models with $d = \{0.1M, 1M, 10M, 100M\}$ distinct samples from the $f_{\mathrm{inv\_sqrt}}$ sampling distribution over a fixed training budget of $b = 100M$ samples. This means that the model sees each distinct sample $b/d$ times during training, so $100M/0.1M = 1000$ times in the $d = 0.1M$ case. We find that as long as the model is exposed to at least $1M$ distinct samples—corresponding to 100 repeats in our experiments—it can readily learn the underlying algorithm (see Table 6).

**Loss regularization prevents model collapse when the task is hard.** Finally, we evaluate how the regularization

*Table 6.* **Seeing some—but not too many—repeated examples aids performance.** We train the models with $q = 257$ and with different numbers of repeated examples (all with the $f_{\text{inv\_sqrt}}$ distribution, angular embedding, and custom loss) and evaluate on the same test set for all. # data repeats is computed as $100M/d$, where $d$ is the number of distinct training examples out of the total $100M$ training budget. $\tau = 0.5\%$ Accuracy is percent of predictions within $0.005q$ of right answer (see §3).

| | $\tau = 0.5\%$ Accuracy | | | |
|---|---|---|---|---|
| # data repeats | $N = 16$ | $N = 32$ | $N = 64$ | $N = 128$ |
| 1000 | 99.3% | 97.7% | 95.7% | 86.1% |
| 100 | **99.8%** | 99.2% | 98.4% | 92.7% |
| 10 | **99.8%** | **99.5%** | **98.9%** | **96.1%** |
| 1 | 99.7% | 99.2% | 97.2% | 91.5% |

term in our custom loss function affects model performance. We train several models with varying $N$ and $q$ on two versions of the loss function from §3.2: one with $\alpha = 10^{-4}$, activating our additional term, and one with $\alpha = 0.0$, which is standard MSE loss. We find that when the task is still easy, regularization is not necessary (Table 7). However, as task difficulty increases, the regularization term increases the probability of success. We explore explanations for this in §5.

*Table 7.* **Model performance with custom and MSE loss are comparable when trained on the modular addition.** We train the models with $q = 257$ and with the best training data parameters identified in §4 with angular embeddings and evaluate on the same test set for all. $\tau = 0.5\%$ Accuracy is percentage of predictions within $0.005q$ of right answer (see §3).

| # Terms $(N)$ | Mod $(q)$ | $\tau = 0.5\%$ Accuracy | |
|---|---|---|---|
| | | Custom Loss $\alpha = 10^{-4}$ | MSE Loss $\alpha = 0$ |
| 16 | 257 | 99.8% | 99.9% |
| | 3329 | 99.7% | 99.7% |
| | 42899 | 99.7% | 99.6% |
| | 974269 | 99.7% | 99.7% |
| 32 | 257 | 99.5% | 99.6% |
| | 3329 | 99.4% | 99.2% |
| | 42899 | 99.4% | 99.3% |
| | 974269 | 99.5% | 98.9% |

**Our approach has minimal computational overhead.** The computational overhead of our approach is minimal compared to the baseline, as we still generate the same number of data samples and simply modify the distribution used for generation. Similarly, the loss regularization term does not introduce any additional cost (besides the negligible cost of calculating the term itself) as we have a standard training loop. We time the difference between the custom loss and standard loss experiments and report that the time difference

is negligible, around 0.04% relative difference.

## 5. Application: Cryptanalysis of LWE

Having established that our methods enable ML models to better learn modular addition, we now apply them to our motivating example: improved ML-enabled attacks on the Learning with Errors (LWE) problem. The goal of these attacks is find $s$ given $A$ and $b$, where $b = A \cdot s + e \pmod q$. $A \in \mathbb{Z}_q^{m \times n}$ is a uniformly random $m \times n$ matrix, the secret $s \in \mathbb{Z}_q^n$ is a vector with length $n$.

For this proof-of-concept evaluation, we ignore the error vector $e$ so we can better study if models learn the harder part of the problem: computing $A \cdot s \pmod q$. If models recover $s$ without noise, recovery with noise should be possible based on experiments from Wenger et al. (2022). We assume $s$ is binary and sparse, based on proposed implementations for LWE in the popular homomorphic encryption setting (Bossuat et al., 2024).

Following Stevens et al. (2024), we train an encoder-only model on $(A, b)$ pairs to predict $b$ from $A$. We use 30 million distinct data examples and a total training budget of 1 billion examples, adding our improvements to the loss function to the training pipeline. We evaluate attack performance by measuring number of successful secret recoveries out of 20 random model initializations. We use the binary distinguisher of Wenger et al. (2022) to recover secrets.

We report the results in Table 8: for $N = 64$ and $q = \{257, 3329, 42899, 974269\}$, the model successfully recovers binary secrets with Hamming weight (number of nonzero elements in the secret) up to 6. This result is significant because Wenger et al. (2024) show that ML attacks on LWE traditionally only work when secret recovery requires effectively summing $\leq 3$ elements mod $q$. Thus, our methods appear to **double** the performance of ML attacks on LWE, paving the way for future study.

**Effect of regularization term in LWE setting.** In the prior section (§4.2), we noted that the regularization term in our loss function aided learning in more difficult problem settings. To explain this, we consider the LW(ithout)E task setting. Without regularization, for a given batch $\mathcal{B}$ with ground truth $(x, y)$ and model predictions $(x', y')$, the MSE loss to optimize is $\ell = \|x - x'\|_2^2 + \|y - y'\|_2^2$. At the beginning of the training, we inspect model predictions, and we observe they are approximately equal, as shown in Appendix E. Using this and given $b = a \cdot s \pmod q$ is uniformly distributed, we conclude that, on average, the model loss $\ell \approx \sum_{t=0}^{q-1} \left( (x' - \cos 2\pi \frac{t}{q})^2 + (y' - \sin 2\pi \frac{t}{q})^2 \right)$. Since $\sum_{t=0}^{q-1} \cos 2\pi \frac{t}{q} = \sum_{t=0}^{q-1} \sin 2\pi \frac{t}{q} = 0$, the minimum loss is achieved when $(x', y') = (0, 0)$. This encourages the model to produce predictions with decreasing magnitude at

each training epoch, ultimately degenerating to predictions centered around the origin. Adding the regularization term encourages the model to avoid the origin, allowing faster and better convergence (see Table 8 for comparison).

We experimented with other alternatives to loss regularization, such as estimating the angle distance between predictions, but found that the regularization term most effectively ensured consistent convergence.

*Table 8.* **Model performs better when trained with our custom loss on the LW(ithout)E problem.** We train the models with the best data parameters from §4 and evaluate if the model successfully recovers the secret. Recovery % is the percent of secrets recovered out of 20 different model initialization.

| # Terms $(N)$ | Hamming weight | Mod $(q)$ | Recovery % | |
|---|---|---|---|---|
| | | | Custom Loss $\alpha = 10^{-2}$ | MSE Loss $\alpha = 0$ |
| 64 | 6 | 257 | **15%** | 0% |
| | | 3329 | **20%** | 0% |
| | | 42899 | **15%** | 0% |
| | | 974269 | **15%** | 0% |
| 128 | 5 | 257 | **10%** | 0% |
| | | 3329 | **15%** | 0% |
| | | 42899 | **15%** | 0% |
| | | 974269 | **10%** | 0% |
| 256 | 4 | 257 | **10%** | 0% |
| | | 3329 | **15%** | 0% |
| | | 42899 | **15%** | 0% |
| | | 974269 | **15%** | 0% |

## 6. Beyond Modular Addition

Finally, we consider whether our methods can aid learning in settings beyond modular addition, including other modular arithmetic tasks and broader functions.

### 6.1. Other Modular Arithmetic Tasks

We investigate whether our methods enable ML models to learn other modular arithmetic functions beyond addition. We train models to predict outputs from these functions, using the same setup as before: encoder-only transformer model with modified data distribution, angular embedding, and custom loss.

**Asymmetric Functions.** Doshi et al. (2024) conjectured that two-layer MLPs can only learn functions that can be represented as $h(g_1(a_1), g_2(a_2), \ldots, g_N(a_N))$ and cannot extend beyond this class. We introduce a class of functions $h : \mathbb{Z}_q^N \to \mathbb{Z}_q$ outside the aforementioned class, where $h_{j,k} = \left(\sum_{i=1}^N a_i^j\right)^2 + a_1^k$, to show that our approach helps models learn other modular arithmetic functions.

For these problems, we add a positional embedding in the transformer since these functions depend on input sequence

positions. Table 9 shows that for $N = 16$ and $q = 257$, we achieve an accuracy exceeding **95%+** for these functions.

*Table 9.* **With our methods, models can learn other modular arithmetic functions with good accuracy ($N = 16, q = 257$).** % Accuracy is percentage of predictions exactly correct.

| Function | $\|$ % Accuracy |
|---|---|
| $h_{j=1,k=1} = (a_1 + \ldots + a_N)^2 + a_1^1 \mod q$ | 95.1% |
| $h_{j=1,k=3} = (a_1 + \ldots + a_N)^2 + a_1^3 \mod q$ | 96.2% |
| $h_{j=2,k=1} = \left(a_1^2 + \ldots + a_N^2\right)^2 + a_1^1 \mod q$ | 95.5% |

**Modular multiplication and scalar product.** We test both modular multiplication and the scalar product of two vectors mod $q$, respectively in Table 10 and Table 11. The angular embedding is designed for addition, so we use standard token embedding in multiplication experiments and compare $f_{\text{inv\_sqrt}}$ to $f_{\text{default}}$. For both tasks, the model with $f_{\text{inv\_sqrt}}$ performs well for smaller $q$, but declines for larger $q$. In all scenarios and in both tasks, we see that the $\tau = 1\%$ accuracy for $f_{\text{default}}$ is around 2%. We also note that the scalar product is more difficult, because the model needs to learn both both multiplications and additions.

*Table 10.* **Our methods also lead to better performance on modular multiplication.** We train the models with the best data parameters from §4. The model performance when trained using $f_{\text{default}}$ is 2% on all different settings. $\tau = 1\%$ Accuracy is percentage of predictions within $0.01q$ of right answer (see §3).

| # Terms $(N)$ | Mod $(q)$ | $\tau = 1\%$ Accuracy |
|---|---|---|
| 16 | 97 | 100% |
| | 257 | 98% |
| | 3329 | 3% |
| 32 | 97 | 100% |
| | 257 | 75% |
| | 3329 | 3% |
| 64 | 97 | 100% |
| | 257 | 65% |
| | 3329 | 3% |

### 6.2. Synthetic Tasks

Finally, we explore whether the data distribution techniques alone aid transformer performance on well-studied synthetic tasks, including:

- Copy task (Graves (2014)): given a vector of size $N$ where each element is sampled from a vocabulary $V$, output an exact copy of the vector. A random guess is correct with probability $V^{-N}$.
- Associative recall task (Graves (2014)): given $N$ keys and $N$ values sampled from two distinct vocabularies

*Table 11.* **Our methods also lead to better performance on the scalar product task.** We train the models with the best data parameters from §4. The model performance when trained using $f_{\text{default}}$ is 2% on all different settings. $\tau = 1\%$ Accuracy is percentage of predictions within $0.01q$ of right answer (see §3).

| # Terms $(N)$ | Mod $(q)$ | $\tau = 1\%$ Accuracy |
|---|---|---|
| 2 | 97 | 100% |
|   | 257 | 100% |
|   | 3329 | 3% |
| 4 | 97 | 100% |
|   | 257 | 30% |
|   | 3329 | 3% |
| 8 | 97 | 78% |
|   | 257 | 2% |
|   | 3329 | 3% |

$V_K$ and $V_V$, retrieve the correct value of one key. A random guess is correct with probability $N^{-1}$.

- Parity task (Abbe et al. (2023)): given a binary vector of size $N$, output the parity of the vector. A random guess is correct with probability $50\%$.

- Selective copy (Gu & Dao): given a vector of size $N$ where $T = 16$ non-zero elements are sampled from vocabulary $V$ and $N - T$ elements are equal to zero, output a copy of the vector, discarding all elements equal to 0.

For these different generative tasks, we trained a decoder-only model with rotary positional embedding ((Black et al., 2022)), 4 layers, and a hidden dimension of 256 on batches of 250 examples for 10M total samples. Similar to Section 3.1, we sample each problem length from a distribution $f$ to vary between 1 and max_length, instead of fixing it to be always max_length. When we use $f_{\text{default}}$ we sample problems with length max_length. During evaluation, we calculate the accuracy on problems with length max_length only. We do not use the custom loss during training, as it was designed for modular arithmetic.

As demonstrated in Table 12, incorporating a mix of easier and harder data can significantly enhance model performance. This is particularly evident in the Associative recall and Parity, where models trained with $f_{\text{default}}$ are essentially making predictions at random.

## 7. Conclusion

This work introduces two key changes to the training process to help ML models learn modular addition. These techniques—varying the diversity of training data and introducing a regularized loss function—enable ML models to add hundreds of elements mod a large $q$ with high accuracy,

*Table 12.* **Training data with more diverse problem lengths yields better accuracy results across different tasks.** Particularly on the Associative recall and Parity, the model fails to learn with $f_{\text{default}}$. % Accuracy is percentage of predictions exactly correct.

| Task | # max_length | % Accuracy | | |
|---|---|---|---|---|
|  |  | $f_{\text{default}}$ | $f_{\text{inv\_sqrt}}$ | $f_{\text{uni}}$ |
| Copy | 32 | **100.0%** | **100.0%** | **100.0%** |
|  | 64 | **100.0%** | **100.0%** | **100.0%** |
|  | 128 | 94.3% | **100.0%** | **100.0%** |
|  | 256 | 81.4% | **98.1%** | 97.4% |
| Associative recall | 8 | 32.5% | **100.0%** | **100.0%** |
|  | 16 | 6.6% | **100.0%** | **100.0%** |
|  | 32 | 3.4% | **100.0%** | **100.0%** |
|  | 64 | 1.8% | **100.0%** | 1.8% |
| Parity | 32 | 50.3% | **100.0%** | **100.0%** |
|  | 64 | 50.6% | 99.8% | **100.0%** |
|  | 128 | 50.0% | **99.7%** | 50.2% |
|  | 256 | 50.2% | **99.4%** | 50.2% |
| Selective copy | 32 | **100.0%** | **100.0%** | **100.0%** |
|  | 64 | **100.0%** | **100.0%** | **100.0%** |
|  | 128 | 83.4% | **100.0%** | **100.0%** |
|  | 256 | 57.2% | **100.0%** | 99.3% |

a significant improvement over prior work. We also demonstrate that these techniques can be applied to the Learning with Errors (LWE) problem in cryptography to train ML models that recover 2x harder secrets than prior work.

Furthermore, our findings provide valuable insights into how models learn modular arithmetic, which can inform the development of more effective ML algorithms for a wide range of tasks. By applying our methods to other well-studied problems, we demonstrated their potential to generalize and improve learning outcomes.

Several interesting directions remain for future work. As the number of terms $N$ increases, our models have slightly higher MSE and lower accuracy. This motivates future work to understand how to improve performance as $N$ scales. Another avenue is more exploration into transferring our techniques to other settings, such as ML-enabled cryptanalysis. While our method achieves success on $q$ used in real cryptosystems and $N$ close to real-world use cases ($N = 512$ is used in practice (Avanzi et al., 2021)) on Learning *without* Errors, more experimentation is needed to transfer modular addition knowledge to Learning *with* Errors with large $N$ and $q$. Possible approaches include pretraining on general modular addition and fine-tuning on specific settings, but future research should consider creative approaches.

## Acknowledgments

The authors wish to thank Zeyuan Allen-Zhu for initially proposing the sampling distribution strategy. We also thank François Charton for discussions on the data repetition effect and Mohamed Malhou and Andrey Gromov for helpful conversations on arithmetic tasks.

## Impact Statement

This paper presents work whose goal is to advance the field of Machine Learning. There are many potential societal consequences of our work, none which we feel must be specifically highlighted here.

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

## A. Additional results

We report additional results using our approach in Table 13 and Table 14 for different non-prime $q$ and in Table 15 for different $N$ that are not powers of 2. We see similar trends as Table 2.

*Table 13.* **Our approach is also successful for non-prime $q = 2^{16}$ and $N \in \{16, 32, 64, 128\}$ elements.** All metrics are computed on a held out test set. MSE is mean squared error, $\tau = 0.5\%$ Accuracy is percentage of predictions within $0.005q$ of right answer, and $\tau = 1\%$ Accuracy is percentage of predictions within $0.01q$ of right answer (see §3 for details).

| # Terms ($N$) | Mod ($q$) | MSE | $\tau = 0.5\%$ Accuracy | $\tau = 1\%$ Accuracy |
|---|---|---|---|---|
| 16 | $2^{16}$ | 0.00 | 99.8% | 100.0% |
| 32 | $2^{16}$ | 0.00 | 99.6% | 100.0% |
| 64 | $2^{16}$ | 0.01 | 98.1% | 99.5% |
| 128 | $2^{16}$ | 0.02 | 95.7% | 98.4% |

*Table 14.* **Our approach also works for different non-prime $q$ and $N \in \{16, 32, 64, 128\}$.** We train the models with the best data parameters from §4. All metrics are computed on a held out test set. $\tau = 1\%$ Accuracy is percentage of predictions within $0.01q$ of right answer (see §3 for details).

| # Terms ($N$) | Mod ($q$) | $\tau = 1\%$ Accuracy |
|---|---|---|
| 16 | 1728 | 100% |
|  | 100000 | 100% |
|  | 1048576 | 100% |
|  | 10000001 | 100% |
| 32 | 1728 | 100% |
|  | 100000 | 100% |
|  | 1048576 | 100% |
|  | 10000001 | 100% |
| 64 | 1728 | 99.5% |
|  | 100000 | 99.3% |
|  | 1048576 | 99.4% |
|  | 10000001 | 99.8% |
| 128 | 1728 | 98.0% |
|  | 100000 | 98.2% |
|  | 1048576 | 98.1% |
|  | 10000001 | 98.8% |

*Table 15.* **Our approach also works with non powers of 2 $N$ and prime $q$.** We train the models with the best data parameters from §4. All metrics are computed on a held out test set. $\tau = 1\%$ Accuracy is percentage of predictions within $0.01q$ of right answer (see §3 for details).

| # Terms ($N$) | Mod ($q$) | $\tau = 1\%$ Accuracy |
|---|---|---|
| 20 | 257 | 100% |
|  | 3329 | 100% |
|  | 42899 | 100% |
|  | 974269 | 100% |
| 49 | 257 | 99.7% |
|  | 3329 | 99.6% |
|  | 42899 | 99.7% |
|  | 974269 | 99.6% |
| 101 | 257 | 98.6% |
|  | 3329 | 98.8% |
|  | 42899 | 98.9% |
|  | 974269 | 98.5% |

We also see that our method is not fully robust to high $N$, but perhaps a longer training time or larger model is needed for higher $N$. We experiment both with a longer training time and a larger model and see that both help with performance on high $N$. We present these results in Tables 16, 17, and 18.

## B. Sample Efficiency Comparison

We investigate why our method succeeds and find that our sampling technique allows for a linear sample complexity, while $f_{\text{default}}$ needs an exponential sample complexity to tackle the problem. This helps to explain why our proposed sampling strategy is so effective.

In Table 19, we measure the number of samples needed to get $< 0.005$ loss and 90% test accuracy.

*Table 16.* **With our approach, performance declines with higher $N$.** All metrics are computed on a held out test set. MSE is mean squared error and $\tau = 1\%$ Accuracy is percentage of predictions within $0.01q$ of right answer (see §3 for details).

| # Terms ($N$) | Mod ($q$) | MSE Loss | $\tau = 1\%$ Accuracy |
|---|---|---|---|
| 256 | 257 | 0.15 | 90.4% |
| 256 | 3329 | 0.14 | 92.7% |
| 256 | 42899 | 0.18 | 91.2% |
| 256 | 974269 | 0.17 | 90.6% |

*Table 17.* **Training 4x longer improves performance on high $N$.** All metrics are computed on a held out test set. MSE is mean squared error and $\tau = 1\%$ Accuracy is percentage of predictions within $0.01q$ of right answer (see §3 for details).

| # Terms ($N$) | Mod ($q$) | MSE Loss | $\tau = 1\%$ Accuracy |
|---|---|---|---|
| 256 | 257 | 0.08 | 94.8% |
| 256 | 3329 | 0.08 | 95.1% |
| 256 | 42899 | 0.09 | 95.0% |
| 256 | 974269 | 0.10 | 94.5% |

*Table 18.* **A 4x larger model also improves performance on high $N$.** All metrics are computed on a held out test set. MSE is mean squared error and $\tau = 1\%$ Accuracy is percentage of predictions within $0.01q$ of right answer (see §3 for details).

| # Terms ($N$) | Mod ($q$) | MSE Loss | $\tau = 1\%$ Accuracy |
|---|---|---|---|
| 256 | 257 | 0.07 | 96.2% |
| 256 | 3329 | 0.07 | 96.7% |
| 256 | 42899 | 0.08 | 96.1% |
| 256 | 974269 | 0.09 | 95.8% |

*Table 19.* **Number of samples needed to get < 0.005 loss and 90% test accuracy with $q = 3329$.**

| $N$ | $f_{\text{default}}$ | $f_{\text{inv\_sqrt}}$ (with best $f_{\text{default}}$ setting) | $f_{\text{inv\_sqrt}}$ (with our best setting) |
|---|---|---|---|
| 6 | 4.5M | 4.1M | **0.6M** |
| 9 | 7.1M | 1.9M | **0.45M** |
| 12 | 12.85M | 2.6M | **0.95M** |
| 15 | 51.1M | 8.15M | **1.3M** |
| 18 | Never | 9.35M | **1.75M** |

## C. Polynomial Sum Tasks

We additionally run two symbolic polynomial tasks mod $q$: (a) sum $N$ polynomials with degree $\leq K$ and (b) sum 2 polynomials with degree $\leq K$.

For both tasks, we fix $q = 3329$ and we encode the polynomial as $a_m, a_{m-1}, \ldots, a_0$, separate polynomials with <SEP> token, and ask the model to predict the coefficients of the polynomial sum. We add a RoPE positional embedding. We report the results comparing $f_{\text{inv\_sqrt}}$ vs $f_{\text{default}}$.

For all experiments, model can quickly predict the degree of the polynomial sum, i.e. the number of tokens to output before the <EOS> token. Therefore the real difference between the two strategies is predicting the right coefficients. We say that the model found the solution if the maximum error across all coefficients is less than $0.01q$.

Task (a) is closely related to our original modular addition task (with $K = 0$), and we see the success of the distribution $f_{\text{inv\_sqrt}}$. Results are reported in Table 20.

Task (b) can be decomposed into finding the right index to attend to and summing two numbers mod $q$ (which can be completely memorized with $\mathcal{O}(q^2)$ even without angular embedding). We show the results on Table 21.

## D. XE as a MSE-like loss

When selecting an appropriate loss function for our model, we had to choose between a MSE-like loss (regression) or a

Table 20. **On summing $N$ polynomials with degree $\leq K$ task, the $f_{\text{inv\_sqrt}}$ distribution shows good performance.** The standard distribution $f_{\text{default}}$ fails to learn the task.

| $N$ | $K$ | Correct % ($f_{\text{inv\_sqrt}}$) | Correct % ($f_{\text{default}}$) |
|---|---|---|---|
| 16 | 1 | 99.5% | 0% |
| | 4 | 99.2% | 0% |
| | 16 | 99.3% | 0% |
| 64 | 1 | 98.8% | 0% |
| | 4 | 98.7% | 0% |
| | 16 | 98.5% | 0% |

Table 21. **On summing two polynomial with degree $\leq K$ task, the $f_{\text{inv\_sqrt}}$ distribution shows spectacular performance, while $f_{\text{default}}$ fails to learn the task.**

| $K$ | Correct % ($f_{\text{inv\_sqrt}}$) | Correct % ($f_{\text{default}}$) |
|---|---|---|
| 64 | 99.5% | 0% |
| 128 | 99.2% | 0% |

cross-entropy-like (XE) loss (classification). The choice is particularly delicate for arithmetic problems, especially if the model decodes a token from a vocabulary (with XE) or an angular representation (with MSE). Our claim is that, in our particular case, MSE is actually superior compared to XE. In particular let $x \in \mathbb{Z}_q$ be a point, and let the vector $e_i \in \mathbb{Z}_q^N$ where all components are zeros except for the $i$-th component, which is 1. Let $g : \mathbb{Z}_q^N \to Z_q$ be the function the model needs to learn (in particular $g(x) = x \cdot \mathbf{1} \pmod{q}$ for the modular arithmetic task and $g(x) = x \cdot s \pmod{q}$, where the secret $s \in \mathbb{Z}_q^N$ is a binary unknown vector, for the LWE task). The finite difference in the $i$-th direction can be defined as

$$\Delta_i g(x) = g(x + e_i) - g(x).$$

It's easy to show that for these tasks, $\max_{x,i} |\Delta_i g(x)| \leq 1$. This equivalently means that the difference between the function evaluated on closely related inputs remains relatively small. In this case, we observed that a well-trained model using XE loss can correctly approximate the aforementioned $g(x)$.

Table 22. **Models perform better when trained with MSE-like loss.** Models trained on the best settings identified in §4 and evaluated on the same test set for all. $\tau = 0.5\%$ Accuracy is percentage of predictions within $0.005q$ of right answer (see §3)

| # Terms ($N$) | Mod ($q$) | | $\tau = 0.5\%$ Accuracy | | |
|---|---|---|---|---|---|
| | | | Custom Loss $\alpha = 1e - 4$ | MSE Loss $\alpha = 0$ | XE Loss |
| 16 | 257 | | 99.8% | 99.9% | 99.5% |
| | 3329 | | 99.7% | 99.7% | 99.2% |
| | 42899 | | 99.7% | 99.6% | 98.1% |
| | 974269 | | 99.7% | 99.7% | 96.4% |
| 32 | 257 | | 99.5% | 99.6% | 98.5% |
| | 3329 | | 99.4% | 99.2% | 96.8% |
| | 42899 | | 99.4% | 99.3% | 94.3% |
| | 974269 | | 99.5% | 98.9% | 91.0% |

After inspecting the cross-entropy output, we noticed that XE loss learned a bell-shape curve centered at the correct solution with a tiny variance (in the order of $0.01q$) which effectively forces the XE loss into a MSE-like loss. This suggests us to prefer a MSE loss and decode the output as regression and as Table 22 shows, we achieve best results using MSE-like loss.

## E. Visualizing model predictions in LWE setting

We analyze the model's output predictions after 100 steps of the training to validate that, when the task is particularly hard, MSE loss cannot be used without the regularization term. To do this, we pass input sequences to the model and extract their outputs. We plot the outputs, coloring them based on $y = x \cdot s \pmod{q}$ of the input sequence $x$. Figure 3 presents this analysis for three models trained with $N = 64$, $q = 257$ and different settings for the hamming weight (number of nonzero elements in the secret) and loss: (a) hamming weight 3 with MSE Loss; (b) hamming weight 6 with MSE Loss; (c) and hamming weight 6 with Custom Loss.

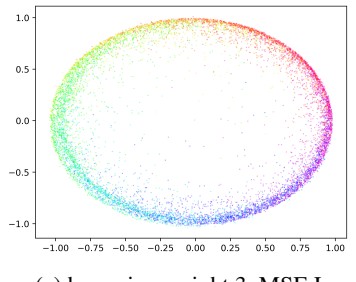
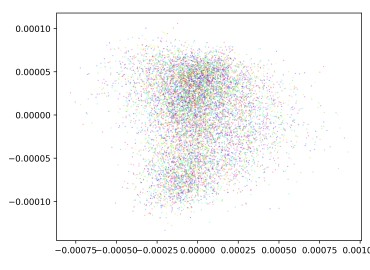
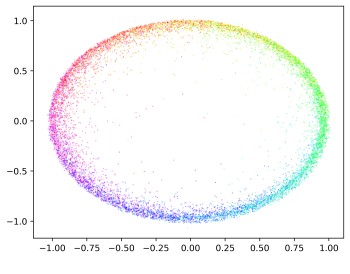

(a) hamming weight 3, MSE Loss      (b) hamming weight 6, MSE Loss      (c) hamming weight 6, Custom Loss

*Figure 3.* **Final model output for different hamming weights and losses.** Plots show the model's final output. Points with the same color have the same output $y = x \cdot s \mod q$ (i.e. they should be close together in representation).

As Figure 3 shows, a model trained with MSE loss can successfully learn the problem when hamming weight is 3. However, for the setting without the regularization term and hamming weight 6, the model fails to learn, and the final output is visually meaningless. Additionally, looking at the magnitude, we empirically verify that predictions for a single batch are tied together and very close to the origin. Once we add the regularization term, the model can learn the task with hamming weight 6. This implies that for small hamming weight, the custom loss is not as important, likely because the problem is simpler, but for larger hamming weight, the custom loss enables learning.

## F. Curriculum Learning Hyperparameter Analysis

We provide the different configurations we tried to get a fair comparison between curriculum learning and our sampling strategy. In bold, we indicate the best setting.

When we ran the CL baselines, we modified three things:

1. Thresholds:

    (a) $T_1$ is either 1% or 3% or 10% of the training
    (b) $T_1$ is when train_loss($X_1$) < eps, where we chose eps = **1e-2**, 1e-3

2. Data mix: we divide the training dataset $X$ into two disjoint sets $X_1$ and $X_2$ where $X_1$ is the subset of $X$ such that each instance contains at least half of the elements equal to zeros.

    (a) Train the model using $X_1$ up to $T_1$, then $X_2$ until the end
    (b) **Train the model using $X_1$ up to $T_1$, then the entire $X$ until the end**

3. Learning rate and weight decay:

    (a) We experimented with 3 choices of learning rate (1e-5, **3e-5**, 1e-4) and 3 choices of weight decay (0.03, **0.1**, 0.3)

