# OpenReview forum: "Making Hard Problems Easier with Custom Data Distributions and Loss Regularization: A Case Study in Modular Arithmetic"
_ICML.cc/2025/Conference — ICML 2025 poster_

### Official Review · Reviewer_Eby5 · 2025-03-12

**Overall Recommendation:** 4

**Summary:**

This paper proposes a new training strategy and loss function for successful modular additions and other operations. The critical observation is the utility of sparse samples. Models learn modular addition better on sparse samples, and if sparse and dense samples are mixed, it learns from sparse samples and then generalizes to dense samples. Previous studies sample training samples uniformly, and this leads the sampling to concentrate on non-sparse elements. The authors proposed to sample numbers from a distribution in which large numbers are sampled less (and thus, more chances of sampling 0). Further, the new loss function includes an additional term that encourages the model's output to be on the unit circle. The idea is based on the observation that in hard settings, the model's output collapses to the origin. In the experiments, the proposed sparsity-aware sampling and the custom loss both are demonstrated to improve the accuracy significantly.

**update after rebuttal.**

I appreciate the author's full elaborations and answers to my concerns, which greatly deepened my understanding of their work. The explanation and additional results address my concerns so I suppose that this work is worth being presented in the main conference.

**Claims And Evidence:**

The claims and evidence are generally reasonable and convincing, but I have a concern that the explanation and the method sometimes assume modular addition, and the authors should include more general discussions.

For example, [l.113] says
> [l.113, left]  Because 0 and q-1 are "close" in a modular field, ...
but this needs several remarks. First, modular field does not equip any distance as it violates the triangle inequality. Thus, being "close" is not precise even with quotations. The two numbers 0 and q-1 might appear "close" because of the unit 1 in the addition, but the field also equips multiplication. Similarly, including many 0 makes problems sparser/easier for addition task, but not necessarily for others. For the multiplication task 1 should be the number that makes the task "sparser."

**Essential References Not Discussed:**

The following paper can be added as the literature of ML applications to hard math problems in [l.37, right]．
1. Learning to compute Gröbner bases, Hiroshi Kera, Yuki Ishihara, Yuta Kambe, Tristan Vaccon, Kazuhiro Yokoyama, NeurIPS'24

This also shows that the polynomial computations over finite-field is unsuccessful due to errors in coefficient predictions. There are also several studies handling easier polynomial tasks (not necessarily working on finite field), and as mentioned above, it should be an interesting future work to see if the proposed method also works successfully on these tasks on finite field.

2. Do Transformers Understand Polynomial Simplification? Vishesh Agarwal, Somak Aditya, Navin Goyal, ICLR'21

**Experimental Designs Or Analyses:**

The experiments carefully examine the claims with variations in the number of terms and modulus. The recent repetition technique is also taken into account, and beyond the modular addition task, LWE cryptanalysis and several standard benchmark tasks are included in the experiments. However, as mentioned in #Methods And Evaluation Criteria, the experiments are strongly based on modular addition. The experiments also include other tasks in the end, but Parity is also an additive task, and the other tasks are not very arithmetic. Including more arithmetic tasks and generalizing the proposed method will make this study even more impactful.

**Methods And Evaluation Criteria:**

The proposed method and evaluation criteria are reasonable. One of the concerns is that the explanation of the proposed method is biased toward modular addition and not general enough. The experiments also put a great focus on the modular addition, and the extended task (Table 8) is also taking additive formats. It would be better for authors to make the explanation more general, and verify them in the experiments.

**Other Comments Or Suggestions:**

The paper is written well in general. As I commented above, the explanation should be carefully revised and make it clear whether it writes about addition or more general cases.

**Other Strengths And Weaknesses:**

Important srengths and weaknesses have been mentioned in other cells. This work is solid and has sufficient contributions to this topic. The proposed methods are simple and easy to adopted.

**Questions For Authors:**

Please refer to the other cells. Particularly, I want a clear explanation of the proposed method beyond addition.

**Relation To Broader Scientific Literature:**

This study has a strong contribution to the modular arithmetic learning. It is interesting to see if the proposed method works for symbolic computation that involves modular arithmetic. For example, symbolic regression over modular field, or polynomial sum, multiplication, reduction, with finite-field coefficients.

**Theoretical Claims:**

No theoretical claims are made.

---

> ### Author Rebuttal · Authors · 2025-04-01
>
> We thank you for your thoughtful feedback.
>
> **Re: claims:** We will include revisions in the final paper to clarify the claims more generally to not assume addition. Thanks for pointing out the clarification on $0$ and $q-1$ in the modular field, we will update this in the final version.
>
> **Re: sparsity:** Regarding choosing to include $0$ as the element for sparsity, we actually found that including a random but fixed number instead of $0$ yields similar results. It is not the number itself but instead the shift in distribution that leads to performance improvements. The key is changing the KL divergence between the train and test sets, which is irrespective of the element chosen for sparsity.
>
> To show this, we ran an additional experiment where we substitute the $0$ with an arbitrary integer $K$ and shift the distribution by sampling more $K$s compared to all remaining elements in $Z_p^N$. We used a random $K$ for each different $q$ to ensure $K$ is not a factor. We can also run with more $K$ to show that this trend persists. We found similar results in the multiplication case.
>
> | $N$ | $q$        | $K$       | tau=1% acc |
> | --- | ---      | ---     | ---       |
> | 32 | 257     | 160    | 100.0% |
> | 32 | 3329    | 3176   | 100.0% |
> | 32 | 42899   | 24606  | 100.0% |
> | 32 | 974269  | 79062  | 100.0% |
> | 64 | 257 | 160    | 99.2% |
> | 64 | 3329 | 3176   | 99.3% |
> | 64 | 42899 | 24606  | 99.4% |
> | 64 | 974269  | 79062  | 99.3% |
> | 128 | 257  | 160 | 97.9% |
> | 128 | 3329 | 3176 | 98.3% |
> | 128 | 42899 | 24606 | 97.9% |
> | 128 | 974269 | 79062 | 97.9% |
>
> **Re: other tasks beyond addition:** We also conducted additional experiments on many other tasks based on your feedback. We present results on the synthetic tasks in our response to reviewer ZgiU and the rest below.
>
> *Modular multiplication and scalar product:* The angular embedding is designed for addition, so we use standard token embedding in multiplication experiments and compare $f_{inv-sqrt}$ to $f_{default}$. We test both modular multiplication and the scalar product of two vectors mod $q$. For both tasks, the model with $f_{inv-sqrt}$ performs well for smaller $q$, but declines for larger $q$. The scalar product is more difficult due to it requiring both multiplications and additions. Still, $f_{default}$ performs around 0% (on acc tau=1%) acc for all settings in both tasks, so $f_{inv\\_sqrt}$ is still an improvement.
>
> **Modular Multiplication with $f_{inv\\_sqrt}$**
> | $N$  | $q$      | tau=1% acc |
> |----|--------|------------|
> | 4  | 97     | 100% |
> | 4  | 257    | 100% |
> | 4  | 3329   | 51% |
> | 8  | 97     | 100% |
> | 8  | 257    | 100% |
> | 8  | 3329   | 32% |
> | 16 | 97     | 100% |
> | 16 | 257    | 98% |
> | 16 | 3329   | 25% |
> | 32 | 97     | 100% |
> | 32 | 257    | 75% |
> | 32 | 3329   | 13% |
> | 64 | 97     | 100% |
> | 64 | 257    | 65% |
> | 64 | 3329   | 3% |
>
> **Scalar Product with $f_{inv\\_sqrt}$**
> | $N$  | $q$ | tau=1% acc |
> |----|--------|------------|
> | 2  | 97  | 100% |
> | 2  | 257 | 100% |
> | 2  | 3329 | 98%  |
> | 4  | 97  | 100% |
> | 4  | 257 | 92% |
> | 4  | 3329 | 83% |
> | 8  | 97 | 78% |
> | 8  | 257 | 38% |
> | 8  | 3329 | 15% |
>
> *Polynomial sum:* We also run on two symbolic polynomial tasks mod $q$: sum $N$ polynomials with degree $\\leq K$ and sum 2 polynomials with degree $\\leq K$.
>
> For both tasks, we encode the polynomial as $a_m, a_{m-1}, …, a_0$, separate polynomials with <SEP> token, and ask the model to predict the coefficients of the polynomial sum. We add a RoPE positional embedding to help the model learn how to count. We fix $q=3329$. We report the results comparing $f_{inv\\_sqrt}$ vs $f_{default}$.
>
> For all experiments, the model is quickly able to predict the degree of the polynomial sum (the number of tokens to output before the <EOS> token), so the real difference between the two strategies is predicting the right coefficients. We say that the model found the solution if the maximum error across all coefficients is less than $0.01q$.
>
> Task a. is closely related to our original task (taking $K=0$)
> | $N$ | $K$ | Correct % for $f_{inv-sqrt}$ | Correct % for $f_{default}$ |
> |----|----|---------------------------|-------------------------------|
> | 16 | 1  | 99.5% | 0% |
> | 16 | 4  | 99.2% | 0% |
> | 16 | 16 | 99.3% | 0% |
> | 64 | 1  | 98.8% | 0%|
> | 64 | 4  | 98.7% | 0% |
> | 64 | 16 | 98.5% | 0%|
>
> Task b.
> | $K$ | Correct % for $f_{inv\\_sqrt}$ | Correct % for $f_{default}$ |
> |----|---------------------------|-------------------------------|
> | 64 | 99.0% | 94.3% |
> | 128| 99.1% | 92.1% |
>
> This task is a bit easier because it can be decomposed into find the right index to attend to (and RoPE is expert on this) + sum two numbers mod $q$ (which can be completely memorized with O($q^2$) even without angular embedding)
>
> **Re: references:** Thanks for sharing the references, we will update the related work section to include these references. We agree that investigating finite-field polynomial tasks is an interesting avenue for future work.

---

> > ### Comment · Reviewer_Eby5 · 2025-04-03
> >
> > Thank you for your answers and for providing additional results. These address my concerns well and I'll retain my positive score on this work.

---

> > > ### Author Response · Authors · 2025-04-07
> > >
> > > Thank you for your helpful feedback and response, we appreciate it!

---

### Official Review · Reviewer_Qesb · 2025-03-14

**Overall Recommendation:** 3

**Summary:**

The paper improves machine learning attack baselines on LWE by training models to do modular arithmetic better. It uses custom training data and a special loss function, allowing the model to sum up to 128 elements modulo q ≤ 974269. It also shows improvements on other tasks like copy, associative recall, and parity.

**Claims And Evidence:**

The paper claims that its methods enable ML models to perform modular addition for up to 128 elements modulo q ≤ 974269—far exceeding prior limits of N ≤ 6 and q ≤ 1000. While the evidence presented supports these claims, it lacks comparisons with other released methods such as the lattice-estimator baseline (https://github.com/malb/lattice-estimator). Without such baselines, it is hard to fully assess the real gains and stability improvements.

**Essential References Not Discussed:**

Unsure

**Experimental Designs Or Analyses:**

The experimental design is generally sound, showcasing improved performance on LWE and other tasks. However, the training process consumes a large amount of data and computation, and a relative comparison with existing approaches is missing. It would strengthen the paper if the authors compared the amortized cost and practical efficiency against established methods. Moreover, while some settings are shown to be easier, it remains unclear whether the method has been evaluated under truly hard conditions. More experiments on challenging settings are needed to validate the robustness of the approach.

**Methods And Evaluation Criteria:**

The methods generally make sense by using distribution aligned with difficulty and better domain knowledge.

**Other Comments Or Suggestions:**

N/A

**Other Strengths And Weaknesses:**

The work is overall promising and sheds light on how modular arithmetic tasks can probe the stability of ML models. Addressing the missing baselines and providing deeper comparisons will further solidify the paper’s contributions.

**Questions For Authors:**

I am confused on how the encoded embedding can map to multiple coefficient? I think the space of embedding would be constrained to decode a large number of terms due to expressiveness?

**Relation To Broader Scientific Literature:**

The technique could help contribute a way for attacking LWE

**Theoretical Claims:**

N/A

---

> ### Author Rebuttal · Authors · 2025-04-01
>
> We thank you for your thoughtful feedback.
>
> **Re: comparisons:** In the paper, we compare our methods to the standard training approach with regular loss and default data distribution for the arithmetic, synthetic and the cryptography tasks (see Tables 3, 6, 7, and 9). We also provide a comparison to curriculum learning for modular addition (Table 4), where we show that our method is more consistent at learning the task compared to the curriculum learning approach. The lattice estimator provides theoretical cost estimates (in bit-operations) for attacking certain instantiations of LWE-based cryptosystems. Since the proof-of-concept we present in this work does not attack actual LWE-based cryptosystems, comparing against the lattice estimator would not be a fair comparison. Future work expanding on this should implement a true LWE setup and can then compare against other LWE attack methods. We welcome any additional suggestions of comparisons we could make in this work.
>
> **Re: cost/efficiency comparison:** We will include an analysis of the cost/efficiency of our method compared to the baselines in the revised version. The computational overhead of our approach is minimal compared to the baseline, as we still generate the same number of data samples and simply modify the distribution used for generation. Similarly, the loss regularization term does not introduce any additional cost (besides the negligible cost of calculating the term itself) as we have a standard training loop. We timed the difference between the custom loss and standard loss experiments and saw that there was a 0.04% difference.
>
> **Re: additional experiments on challenging settings:** As mentioned in our response to reviewers ZgiU and Eby5, we conducted additional experiments on more challenging settings and other tasks, which we will include in the revised version.
>
> We conducted additional experiments on more values of $N$ and $q$, including non powers of 2 and non primes, for modular addition. We find that our method is robust to increased $q$s and non-powers of 2 $N$s.
>
> **Results for Different Values of $N$ and $q$**
> | $N$ | $q$        | tau=1% acc |
> | --- | ---      | ---       |
> | 16 | 1728     | 100.0%    |
> | 16 | 100000   | 100.0%    |
> | 16 | 1048576  | 100.0%    |
> | 16 | 10000001 | 100.0%    |
> | 32 | 1728     | 100.0%    |
> | 32 | 100000   | 100.0%    |
> | 32 | 1048576  | 100.0%    |
> | 32 | 10000001 | 100.0%    |
> | 64 | 1728     | 99.5%     |
> | 64 | 100000   | 99.3%     |
> | 64 | 1048576  | 99.4%     |
> | 64 | 10000001 | 99.8%     |
> | 128 | 1728    | 98.0%     |
> | 128 | 100000  | 98.2%     |
> | 128 | 1048576 | 98.1%     |
> | 128 | 10000001| 98.8%     |
>
> **Results for Different Values of $N$ (non-powers of 2) and $q$**
> | $N$ | $q$        | tau=1% acc |
> | --- | ---      | ---       |
> | 20 | 257     | 100.0% |
> | 20 | 3329    | 100.0% |
> | 20 | 42899   | 100.0% |
> | 20 | 974269  | 100.0% |
> | 49 | 257     | 99.7%     |
> | 49 | 3329    | 99.6%     |
> | 49 | 42899   | 99.7%     |
> | 49 | 974269  | 99.6%     |
> | 101 | 257    | 98.6%     |
> | 101 | 3329   | 98.8%     |
> | 101 | 42899  | 98.9%     |
> | 101 | 974269 | 98.5%     |
>
> We also see that our method is not fully robust to high $N$, but perhaps a longer training time or larger model is needed for higher $N$.
>
> | $N$ | $q$        | MSE loss | tau=1% acc |
> | --- | ---      | ---       | ---       |
> | 256 | 257 | 0.15  | 90.4% |
> | 256 | 3329 | 0.14 | 92.7%  |
> | 256 | 42899 | 0.18 | 91.2% |
> | 256 | 974269 | 0.17 | 90.6% |
>
> **1. What happens if you train 4x longer?**
> | $N$ | $q$        | MSE loss | tau=1% acc |
> | --- | ---      | ---       | ---       |
> | 256 | 257 | 0.08 | 94.8% |
> | 256 | 3329 | 0.08 | 95.1% |
> | 256 | 42899 | 0.09 | 95.0% |
> | 256 | 974269 | 0.10 | 94.5% |
>
> **2. What happens if your model is 4x larger (embed dim goes from 256 to 512)?**
> | $N$ | $q$        | MSE loss | tau=1% acc |
> | --- | ---      | ---       | ---       |
> | 256 | 257 | 0.07 | 96.2%  |
> | 256 | 3329 | 0.07 | 96.7%  |
> | 256 | 42899 | 0.08 | 96.1% |
> | 256 | 974269 | 0.09 | 95.8% |
>
> We also conducted experiments on additional arithmetic tasks (product of n numbers and scalar product of two vectors) and synthetic tasks (multi hop question answering and selective copy). These results are presented in our response to reviewer Eby5 (“other tasks beyond addition”). We find that our method is robust to these additional tasks. Specifically, we vary the distribution of the input length with our custom distributions, and we find that this still leads to performance improvements on these additional tasks. We are also happy to experiment with any other tasks the reviewers suggest.
>
> **Re: embedding question:** Thanks for the question. We use the angular embedding from Stevens et al. (2024) in our work. Could you please clarify your question on the embedding? We want to address your question appropriately but didn’t quite understand it well enough to answer.

---

### Official Review · Reviewer_ZgiU · 2025-03-14

**Overall Recommendation:** 4

**Summary:**

The paper addresses the challenge machine learning models face in learning modular arithmetic, specifically in the context of the Learning with Errors (LWE) problem. It proposes two techniques: (i) using a designed data distribution that mixes sparse and dense modular arithmetic instances, and (ii) introducing a custom loss function with angular embedding and regularization to discourage convergence to trivial local minima. Experimental evaluation demonstrates an increase in performance of ML models on modular arithmetic task.

**Claims And Evidence:**

Overall, the paper presents convincing empirical evidence supporting the claimed improvements. Experiments clearly demonstrate that custom data distributions and the novel loss function substantially enhance accuracy across various problem complexities. However, the claim regarding generalization to a other set of arithmetic and synthetic tasks, while promising, could benefit from further detailed experiments and additional baseline comparisons (for example check other N values).

**Essential References Not Discussed:**

Mostly discussed, however this area is not my primary area, I may have missed some references.

**Experimental Designs Or Analyses:**

The experimental evaluation is sound, and the experiments are comprehensive (different q, number of terms N). However, there is limited hyperparameter sensitivity analysis. Specifically, it needs a more detailed and explicit sensitivity analysis for the improvements over CL.

**Methods And Evaluation Criteria:**

The proposed methods (custom data distribution sampling and the custom regularization of the loss function) are well-motivated by observations from prior literature and empirical insights. The evaluation criteria, including mean squared error (MSE) and accuracy metrics, are standard for the problem context.

**Other Comments Or Suggestions:**

* It would be beneficial to discuss the computational overhead introduced by these modifications compared to the baseline approach.

**Other Strengths And Weaknesses:**

Strengths:
* The paper is well written and structured
* The experimental evaluation is sound.
* Demonstrated significant improvements over SOTA works.

Weaknesses:
* Lack of detailed theoretical explanation for observed improvements limits understanding of the underlying mechanisms.
* Experiments do not fully address the practical cryptographic setting (i.e., noisy LWE instances).
* Generalization experiments, while promising, are limited and do not fully substantiate claims of broader applicability.

**Questions For Authors:**

I do not have any specific questions for the authors.

**Relation To Broader Scientific Literature:**

I think the paper has sufficient novelty, the authors make two contributions which 1) Augmenting Training Data with Sparse Vectors. 2)  Loss Regularization to Avoid Model Collapse. Both of which in relation to prior work are new contributions.

**Theoretical Claims:**

The paper does not propose new theoretical claims requiring formal proofs and builds on theoretical insights from previous works.

---

> ### Author Rebuttal · Authors · 2025-04-01
>
> We thank you for your thoughtful feedback.
>
> **Re: limited generalization experiments:** Per your suggestion, we conducted additional experiments with more $N$ and $q$ values. These results are presented in the response to reviewer Qesb (“additional experiments on challenging settings”).
>
> We also conducted experiments on additional arithmetic tasks (product of n numbers, scalar product of two vectors, polynomial sum) and synthetic tasks (multi hop question answering and selective copy). We present the results for the synthetic tasks below and present the other results in our response to reviewer Eby5 (“other tasks beyond addition”).
>
> *Multi hop Question Answering:* We synthetically implement the [associative recall](https://arxiv.org/pdf/2212.14052) with multi hop. Given $N$ pairs $(a_i, b_i)$ which represent a random permutation from $\\{1, 2, …, N\\}$ to $\\{1, 2, …, N\\}$, we want to find the 2-th successor, i.e. $\\sigma(\\sigma(x))$.
>
> | # max_length | layers | $f_{default}$ | $f_{inv\\_sqrt}$ | $f_{uni}$ |
> | --- | --- | --- | --- | --- |
> | 16 | 4 | 7% | 100% | 100% |
> | 32 | 8 | 3% | 96% | 99% |
> | 64 | 12 | 2% | 93% | 94% |
> | 128 | 12 | 1% | 91% | 90% |
>
> *Selective copy:* Given a vector of size $N$ where each element is sampled from vocabulary $V$, output a selective copy of the vector (all tokens different from the <JUNK> token). This task was introduced by [Mamba paper](https://arxiv.org/pdf/2312.00752).
>
> | # max_length | $f_{default}$ | $f_{inv\\_sqrt}$ | $f_{uni}$ |
> | --- | --- | --- | --- |
> | 32 | 100% | 100% | 100% |
> | 64 | 100% | 100% | 100% |
> | 128 | 83% | 100% | 100% |
> | 256 | 57% | 100% | 99% |
>
> We find that our method is robust to these additional tasks. Specifically, we vary the distribution of the input length with our custom distributions, and we find that this still leads to performance improvements on these additional tasks. We are also happy to experiment with any other tasks the reviewers suggest.
>
> We will include these additional results and analysis in the revised version.
>
> **Re: hyperparameter sensitivity:** We will include more details on the hyperparameter sensitivity analysis in the revised version. While modifying certain parameters in the curriculum can slightly improve performance, the CL approach is much more involved and requires more tuning to the specific task. Our approach is simpler and provides consistent improvement over sampling from the default distribution. We provide the specific parameters here for your reference.
>
> $X_1$ = at least half of the elements are zeros, $X_2$ = at maximum half of the elements are zeros. \
> When we ran the CL baselines, we modified three things:
> 1. data mix: \
>             i) Using $X _1$ up to $T_1$, then $X_2$ until the end \
>             ii) **Using $X_1$ up to $T_1$, then $X_1$ union $X_2$ until the end**
> 2. Thresholds: \
>             i) $T_1$ is either 1% or 3% or 10% of the training \
>             ii) $T_1$ is when train_loss($X_1$) < eps, where we chose eps = {**1e-2**, 1e-3}
> 3. lr and weight decay: \
>             i) We experimented with 3 choices of lr (1e-5, **3e-5**, 1e-4) and 3 choices of weight decay (from 0.03, **0.1**, 0.3)
>
> In table 4 of the paper we reported the best choice, which is in bold above.
>
> **Re: weaknesses**
> * We are happy to include more explanation on the observed improvements. We did investigate why our method succeeds and found that our sampling technique allows for a linear sample complexity, while $f_{default}$ needs an exponential sample complexity to tackle the problem. This helps to explain why our proposed sampling strategy is so effective.
>
> Below, we measure the number of samples needed to get <0.005 loss and 90% test accuracy.
>
> | $N$ | $f_{default}$ | $f_{inv-sqrt}$ (with best $f_{default}$ setting) | $f_{inv\\_sqrt}$ (with our best setting) |
> | --- | --- | --- | --- |
> | 6   | 4.5M | 4.1M   | 0.6M  |
> | 9   | 7.1M | 1.9M | 0.45M |
> | 12  | 12.85M | 2.6M | 0.95M |
> | 15  | 51.1M | 8.15M  | 1.3M |
> | 18  | Never | 9.35M | 1.75M |
>
> * Experimenting on the full practical cryptographic setting is an important area for future work, and this work provides the foundation for improving performance on practical settings.
> * See above for response to generalization experiment limitations
>
> **Re: computational overhead:** The computational overhead of our approach is minimal compared to the baseline, as we still generate the same number of data samples and simply modify the distribution used for generation. Similarly, the loss regularization term does not introduce any additional cost (besides the negligible cost of calculating the term itself) as we have a standard training loop. We timed the difference between the custom loss and standard loss experiments and saw that there was a 0.04% difference. We will include this explanation in the revised version of the paper.

---

> > ### Comment · Reviewer_ZgiU · 2025-04-03
> >
> > Thank you for your response, it answers the points raised in the review, I am increasing my score.

---

> > > ### Author Response · Authors · 2025-04-07
> > >
> > > Thank you for your helpful feedback and response, we appreciate it!

---

### Decision · Program_Chairs · 2025-05-01

**Decision:**

Accept (poster)

**Comment:**

This work develops machine learning techniques that achieve enhanced performance for modular arithmetic. A key application is developing stronger ML attacks against the Learning with Errors (LWE) problem. The methods are evaluated experimentally and exhibit
 increased performance over prior work. Overall, the reviewers agreed that this is a solid contribution.